# Mapping Compound Databases to Disease Maps—A MINERVA Plugin for CandActBase

**DOI:** 10.3390/jpm11111072

**Published:** 2021-10-24

**Authors:** Liza Vinhoven, Malte Voskamp, Manuel Manfred Nietert

**Affiliations:** 1Department of Medical Bioinformatics, University Medical Center Göttingen, Goldschmidtstraße 1, 37077 Göttingen, Germany; liza.vinhoven@med.uni-goettingen.de (L.V.); malte.voskamp@stud.uni-goettingen.de (M.V.); 2CIDAS Campus Institute Data Science, Goldschmidtstraße 1, 37077 Göttingen, Germany

**Keywords:** systems medicine, disease maps, drug targets, drug repurposing, knowledge repository, data integration

## Abstract

The MINERVA platform is currently the most widely used platform for visualizing and providing access to disease maps. Disease maps are systems biological maps of molecular interactions relevant in a certain disease context, where they can be used to support drug discovery. For this purpose, we extended MINERVA’s own drug and chemical search using the MINERVA plugin starter kit. We developed a plugin to provide a linkage between disease maps in MINERVA and application-specific databases of candidate therapeutics. The plugin has three main functionalities; one shows all the targets of all the compounds in the database, the second is a compound-based search to highlight targets of specific compounds, and the third can be used to find compounds that affect a certain target. As a use case, we applied the plugin to link a disease map and compound database we previously established in the context of cystic fibrosis and, herein, point out possible issues and difficulties. The plugin is publicly available on GitLab; the use-case application to cystic fibrosis, connecting disease maps and the compound database CandActCFTR, is available online.

## 1. Introduction

One of the main aims of systems biology and systems medicine is to understand and model molecular mechanisms in diseases, which can support the development of novel therapeutics. For this purpose, disease maps are being developed to represent existing knowledge on disease mechanisms in a computationally readable and comprehensive manner [1]. These disease maps can then be used by clinicians and experimental scientists as well as computational scientists for different purposes, such as structuring high-throughput data, identifying disease biomarkers, developing better diagnostics and also identifying potential drug targets and drug repositioning [1,2].

To make disease maps publicly available in a comprehensive way, an open source visualization platform for disease maps and molecular interaction networks was developed in 2016 under the name of Molecular Interaction Network Visualization (MINERVA) [3]. MINERVA provides the means to visualize disease maps and make them accessible and explorable for the public, as well as individualize and extend the platform through plugins to suit one’s purposes. As described above, disease maps are especially useful for structuring high-throughput data and assisting in the identification of drug targets and potential therapeutics [1]. A common method in the field of drug discovery is high-throughput screens, where hundreds of small molecules are tested in a certain disease context. These often lead to promising lead substances; however, the means by which the compounds achieve their effects remain unclear in most cases. In this context, disease maps can be used to elucidate the mechanism of action of potential therapeutics that have been tested in high-throughput screens and also identify potential side-effects or adverse reactions that cannot be detected during the screening process.

For this purpose, this project aimed to create a MINERVA plugin to link disease maps and application-specific databases for drug candidates from the literature and high-throughput screens. It provides the means to highlight targets of promising compounds, but also to search for compounds that target a specific protein in the disease map. As a use case, we applied the plugin to link our compound database CandActCFTR (https://candactcftr.ams.med.uni-goettingen.de/; last accessed 23 October 2021) and our CFTR Lifecycle Map [4], which aim at supporting drug discovery and data structuring in cystic fibrosis research.

The MINERVA platform [3] comes with its own drug and chemical searches, which can be found directly on a tab in the user panel. MINERVA’s own drug and chemical searches provide an interface to the DrugBank [5], ChEMBL [6] and the Comparative Toxicogenomics Database (CTD) [7]. Here, the user can search for drugs or chemicals, and different relevant databases will be queried for known targets in the map, which are then highlighted by a pin and displayed in the panel. The drug and chemical searches are separated from each other and use differently shaped pins for differentiation between different compounds. Whereas drugs can be searched by name or brand name and the databases DrugBank [5] and ChEMBL [6] will be queried, chemicals can be searched via their synonyms, querying the CTD [7]. The chemical search only displays interactions with direct evidence for the respective disease. Potential drugs that have not been proven confirmed in this particular disease context are not shown, which restricts the plugin’s use for, for example, drug-repurposing applications.

Furthermore, in 2019, the MINERVA *Drug reactions* plugin [8] was developed, which is based on MINERVA’s own drug search and aims at exploring adverse reactions of drugs that are interacting with entities in a given disease map. It connects to an external data file [9] source and uses MINERVA’s drug search to find the targets of any of the drugs in the database map. The results are displayed as a table in the panel, which shows the drug, the entity with which it interacts and any known drug reactions, such as adverse reactions, warnings and precautions. Additionally, all the targets are highlighted by a pin in the map.

While the available plugins described above allow querying for targets of specific drugs and chemicals, they do not support the reverse case, i.e., the search for drugs or chemicals that interact with a specific target in the map. Moreover, our plugin was developed to integrate data from custom databases of candidate therapeutics. This makes it possible for researchers to integrate their own experimental data or compound collections from different sources for a specific use case. One advantage of using custom databases instead of or on top of generic databases such as PubChem [10] is that all the compounds tested for a specific disease are bundled in one place and information is stored and presented in a way and format that are convenient for its users. Furthermore, if compounds were synthesized specifically for testing in a study, they often cannot be found in generic databases.

Mapping custom collections of candidate compounds to disease maps can provide a means for elucidating drug mechanisms. This is especially important in light of high-throughput drug screens, where hundreds or thousands of compounds are tested in certain settings, but the mechanisms of action of promising lead substances often remain unclear. It can, therefore, be useful to display and query whole custom lists of chemicals that have been tested with regard to a specific disease context in a disease map in MINERVA. The current MINERVA plugins do not allow for the query of large custom datasets, which would be a valuable application for making MINERVA more utilizable for the interpretation of one’s own experimental data from high-throughput screens. Furthermore, it can also be useful to search for compounds that affect a certain target in the disease map, a functionality not yet supported by the available MINERVA plugins. We previously developed the generic IT solution CandActBase [11] for the collection and organization of data on drug candidates tested in a certain context (manuscript submitted for publication). The MINERVA-CandActBase plugin shown here offers a linkage between these application-specific databases of potential drug candidates and pathway data encoded as disease maps on the MINERVA platform [3]. For this purpose, we extended the data from the CandActBase database with data on compound–gene interactions from the ChEMBL database [6] and the Comparative Toxicogenomics Database (CTD) [7].

## 2. Materials and Methods

### 2.1. Implementation

The prototype of our tool was implemented as a MINERVA plugin that runs on the same tomcat instance as the MINVERVA web app [3]. From there, it provides an interface to the databases via their web APIs [6,7,11].

The concept and data flow of the plugin are shown in Figure 1. In order to identify interactions between the compounds in the database and genes and gene products in the disease map, in a first step, the properties of each compound in CandActBase were extracted from PubChem [10] via the PubChem CID. These data were then filtered to obtain the names of the targets connected to the respective compounds from the ChEMBL Protein Target Tree within PubChem. These target names were then searched directly in the ChEMBL database, where the official names of corresponding genes for searched targets can be found. Secondly, the CTD [7] was called via its API to collect more compound–gene interactions. To identify the compounds from the CandActBase correctly, the CAS registry number from the already downloaded PubChem data was used as a query. With the CAS number, we could then call the CTD API and extract chemical–gene interactions for each compound. The resulting data from both databases were then restructured, so they can be compared and filtered for overlaps with the given disease map. The data are stored in the easy-to-use and storage-saving JSON format directly on the hosting server to ensure fast loading times. The first JSON file consists of the compound identifier used in the CandActBase and the name of the genes targeted by each compound. Conversely, the second JSON file includes every gene or protein name from the graph and a list of compound IDs that target the given gene or protein.

The code structure of the plugin is based on the existing MINERVA plugin starter kit (https://git-r3lab.uni.lu/minerva/plugins/starter-kit; last accessed 18 October 2021) [8]. We adjusted the existing methods to fit our purposes, as well as writing new methods to expand the functionality.

We designed our graphical user interface (GUI) to be well structured and visually matching with the color scheme of the CandActCFTR website (https://candactcftr.ams.med.uni-goettingen.de/; last accessed 23 October 2021) as shown in Figure 2 with an exemplary search for the compound “Curcumin” and the identified targets in the graph.

### 2.2. Exemplary Queries

Figure 2 shows the graphical user interface of the plugin (Figure 2b) integrated into the CFTR-disease map hosted on MINERVA (Figure 2a). In the following, the functionalities of the plugin are explained via different queries.

#### 2.2.1. Compound Search

As an example, we searched for the compound “Curcumin”, which is typed into the search bar at the top of the plugin on the right side of the window. The connected database, in this case, CandActCFTR, is then queried for the search term. If a match is found, the search results are then displayed in the box below the search bar, where the ID, the InChIKey, the SMILES Code, the chemical structure and synonyms of the relevant compounds are shown (1). By clicking the “Show Targets” button below a search result, the targets of the desired compound can be highlighted in the disease map (A). The currently selected compounds can be seen in the status bar at the bottom of the plugin interface, which also displays the number of targets highlighted in the disease map.

#### 2.2.2. Compare Two Compounds

To search for common targets of two compounds, the button “Compare Compound” under the initial compound in question can be clicked; a second compound can then be selected, and the common targets are highlighted in the disease map. Furthermore, a list of all the highlighted targets in the graph can be called by clicking the “Show list of targets” button (4).

#### 2.2.3. Reverse Target Search

In the reverse search, any entity in the disease map can be used by selecting a target in the disease map and clicking on the “Select one element in graph” button in the lower half of the plugin interface (2). As a result, all compounds found to have an interaction with the chosen entity are listed in the plugin interface on the side in a style similar to the synonym search results (1).

#### 2.2.4. Show Database-Disease Map Coverage

In order to show the database coverage on the disease map, all targets from all compounds in the CandActBase database can be highlighted in the graph (3).

The plugin application connecting CandActCFTR and the disease maps is available at https://cf-map.uni-goettingen.de (last accessed 23 October 2021); an independent version can be downloaded from GitLab (Appendix A) and used in any MINERVA instance.

## 3. Results

We developed a MINEVRA plugin to link disease maps and application-specific compound databases, using the CandActCFTR database designed for cystic fibrosis as a use case. Our plugin offers three main functionalities:

The show all function allows the user to highlight all the genes and gene products in the disease map that are targeted by one or more chemicals in the custom database. This makes it possible to see which compartments and pathways of the disease map are already covered by potential drugs.

The compound search allows the user to search a specific compound in the compound database by name and synonym, and will be extended to also search by unique identifiers such as SMILES, InChIKey or PubChem CID. The database is then queried for matching compounds, and, upon selection, the targets deposited in the database for the individual compound are highlighted in the map and displayed in the plugin panel. Additionally, the database can be queried for two compounds, and the entities targeted by both compounds are highlighted and listed, which allows the user to identify similarities by inspecting the target overlaps and thus compare the sites of action for different chemicals.

Furthermore, the target search offers the reverse of the compound search, where the user can select a gene or protein from the displayed disease map and the compound database is searched for all the compounds that target this specific entity. This allows users to explore specific pathways and targets in the disease map with regard to potential therapeutics.

## 4. Discussion

In recent years, great efforts have gone into developing systems biology models detailing the molecular, cellular and physiological interactions for different diseases. Not only do disease maps represent current knowledge in a comprehensive manner, but they can also be used to identify drug targets, propose potential drugs, elucidate the mechanisms of action for active compounds and detect possible adverse effects. In order to support these endeavors, we have created a plugin linking disease maps displayed on the well-established MINERVA platform and application-specific compound databases, such as ones based on the CandActBase database solution.

The main advantage of using custom compound databases over generic ones is that the respective data structure and information on each compound can be tailored to the users and the use case in question. Furthermore, all the compounds are available from one place, and it is guaranteed that all the compounds have an entry, which is not always the case in generic databases, for example, due to newly synthesized compounds.

Using cystic fibrosis as a use case, we, here, linked our CF disease map and CandActCFTR, our database for compounds tested as cystic fibrosis therapeutics. The plugin uses gene–chemical interactions parsed from ChEMBL and the CTD to map compounds from the database to the disease map, which is displayed in MINERVA. We implemented three main functions: the first to show all the targets of all the compounds in the database, the second to retrieve the targets of a specific compound, and the third to retrieve all the compounds interacting with a specific target.

In order to ensure fast and responsive loading times, even when handling chemical–gene interactions from multiple queries, we decided to download the interaction data from the databases beforehand. Having the interaction data in local files instead of parsing them directly from the databases also has the advantage that one’s own interaction data can be used or added. The downside of such a system is the lack of recentness and completeness. This could be overcome by creating a refreshing script that downloads, formats and updates changes in entries received from the called databases. Thus, integrating the goal of fast responsiveness in the system, as well as integrating the latest data, would be achieved.

For our use case, only entirely freely accessible databases were used to adhere to the project’s open-source notion. Here, ChEMBL [6] and CTD [7] were used as example databases. Data from other gene–compound interaction databases or own experiments can readily be included for other use cases. Our plugin is easily adaptable to other databases, if their data can be transformed to the common JSON data format. Furthermore, compounds as well as targets should be stored using a common unique identifier, such as the official gene symbols for targets and the InChIKey for compounds.

The lack of consistent identifiers and the huge variety in the data structure of different databases were the biggest hurdles in this project. There is no standardized data format on which to rely. Even basic information, such as common and unique identifiers for chemicals, differs across databases. This is naturally due to the variable use cases of databases, but it would simplify the development of cross-database applications and interdisciplinary research endeavors considerably if standardized structures and identifiers existed.

## 5. Conclusions

We were able to visualize chemical–gene interactions on the MINERVA platform and built a plugin that serves as a template for connecting new, customized database sources to be used in this interactive visualization solution. We thus extended the functionality of the MINERVA platform and showed what a customized plugin could look like.

It is now also possible for clinicians and biologists to view and compare chemicals from the CandActCFTR project with respect to the genes or proteins with which they interact. Additionally, possible side effects resulting from mutual targets of drug combinations can be taken into account based on previous research. The tool can also be applied to drug-repurposing efforts by searching for possible targets of known drugs in different disease maps and, conversely, by searching for drugs through specific targets in a disease map.

The tool can be applied in other disease contexts and to other disease maps. The code and our example data are available on GitLab with instructions on how to implement it for new disease maps (https://s.gwdg.de/SU0rqd, accessed on 23 October 2021).

## Figures and Tables

**Figure 1 jpm-11-01072-f001:**
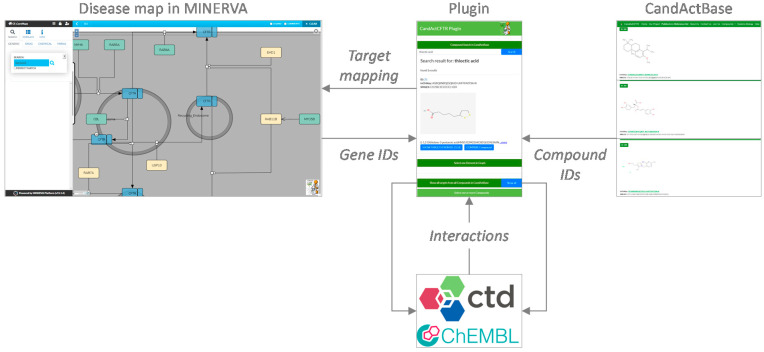
Concept and data flow of the plugin. Gene IDs and Chemical IDs are extracted from the disease map on MINERVA [3] (© Université du Luxembourg, 2021) and the CandActCFTR database, respectively. Using these unique identifiers, gene-chemical interactions are retrieved from the databases ChEMBL [6] (© EMBL-EBI 2018) and CTD [7] (© 2012–2021 NC State University) via the plugin. The results are then listed in the plugin and displayed in the disease map on MINERVA.

**Figure 2 jpm-11-01072-f002:**
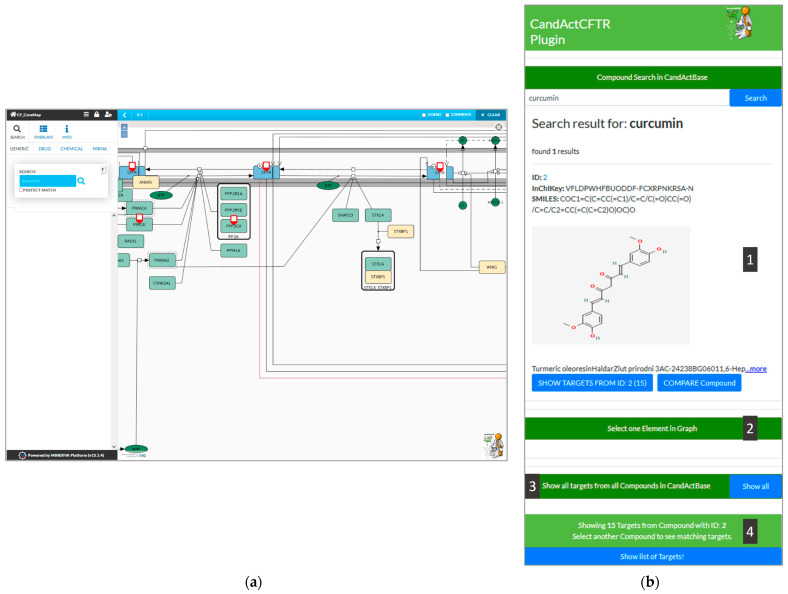
Screenshot from the MINERVA instance showing (**a**) a disease map and highlighted targets, and (**b**) a zoomed-in feature of the UI of the CandActBase plugin. The plugin interface allows the use of all functions from one window: 1. display results from compound query (ID, InChIKey, SMILES Code, chemical structure and synonyms), 2. reverse search by target from disease map, 3. button to highlight all targets of all database compounds in the disease map and 4. button to show targets of selected compound as list.

## Data Availability

Publicly available datasets were analyzed in this study. This data can be found here: https://candactcftr.ams.med.uni-goettingen.de/, https://www.ebi.ac.uk/chembl/, http://ctdbase.org/, accessed on 23 October 2021.

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
