# Peer review of "Mapping Compound Databases to Disease Maps—A MINERVA Plugin for CandActBase"

_jpm, 2021, doi:10.3390/jpm11111072_

Round 1
Reviewer 1 Report
The authors described their development of a useful plugin for the MINERVA platform. Perhaps the value of such tool is their link to specific databases of candidate therapeutics. However this point is not clearly demonstrated in this manuscript. Since MINERVA has its own drug and chemical search tool, the authors should prove why the link to drug candidates tested in a certain context such as their “CandActBase” is important and necessary. Otherwise, users may think MINERVA is enough since computational scale and reverse search functionality could be amended easily. Besides, the authors are suggested to simplify and revise some parts of the writing such as Line 17, Line75-90, discussion and conclusion.
Author Response
The authors described their development of a useful plugin for the MINERVA platform.
We thank the reviewer for their kind assessment of our plugin.
Perhaps the value of such tool is their link to specific databases of candidate therapeutics. However this point is not clearly demonstrated in this manuscript. Since MINERVA has its own drug and chemical search tool, the authors should prove why the link to drug candidates tested in a certain context such as their “CandActBase” is important and necessary. Otherwise, users may think MINERVA is enough since computational scale and reverse search functionality could be amended easily.
We thank the reviewer for pointing out that the added value of our tool has not come across properly. We now added paragraphs explaining the advantages of linking specific databases of candidate therapeutics. The information stored in custom databases and the format this is done in can be tailored to the specific use cases and users. Furthermore, some compounds, e.g. newly synthesized ones, can be missing from generic compound databases such as PubChem. Having all relevant information on all tested compounds in one place and mapping its contents to the disease maps in MINERVA therefore ensures usability, relevance and completeness.
Besides, the authors are suggested to simplify and revise some parts of the writing such as Line 17, Line75-90, discussion and conclusion.
We thank the reviewer for their suggestion. We have revised and rephrased the discussion and conclusion, as well as the parts of writing that correspond to lines 17 and 75-90 in the original manuscript. It is now clearer and better understandable.
Reviewer 2 Report
In the presented study, Vinhoven et al. developed a plugin in the MINERVA platform to map compound databases with disease maps. This plugin links targets with compound databases and allows users to find compounds for all the targets in the disease map and vice versa.
Below are my comments:
- “While the available plugins allow to query for targets of specific drugs and chemicals, they do not support the reverse case, i.e., the search for drugs or chemicals that interact with a specific target in the map” What are these plugins? The author should show detailed comparisons for these plugins. What databases were used in existing plugins with similar functionalities?
- Why did the author choose ChEMBL and CTD and not include DrugBank and other drug and compound databases?
- The authors explained the use case of CandActCFTR. It is not clear how the developed plugin will be implemented with new disease maps.
- The installation instructions for the plugin are not clear.
- The author mentions that the developed plugin is freely available in GitLab, but the GitLab link was not included in the manuscript. The only GitLab link provided is of starter-kit.
- Is the presented plugin allow the mapping of user-provided data? If yes, what type of data and how to upload this. Are there additional sources available to do that?
Author Response
In the presented study, Vinhoven et al. developed a plugin in the MINERVA platform to map compound databases with disease maps. This plugin links targets with compound databases and allows users to find compounds for all the targets in the disease map and vice versa.
We would like to thank the reviewer for their assessment of our manuscript.
Below are my comments:
1. “While the available plugins allow to query for targets of specific drugs and chemicals, they do not support the reverse case, i.e., the search for drugs or chemicals that interact with a specific target in the map” What are these plugins? The author should show detailed comparisons for these plugins. What databases were used in existing plugins with similar functionalities?
We have added information to the paragraphs describing the other plugins in the introduction to compare them better with our own plugin. MINERVAs own drug and chemical search uses DrugBank, ChEMBL and the CTD, but limits the latter to compounds with a direct evidence to the respective disease. The second plugin (Drug Reactions) shows adverse effects between two compounds and uses an external data file based on data from Demner-Fushman et al., 2018.
2. Why did the author choose ChEMBL and CTD and not include DrugBank and other drug and compound databases?
We thank the reviewer for pointing this out. We have added a paragraph explaining our choice in compound databases to the discussion. In the notion of our project, we put a high emphasis on open source, so we decided not to use and redistribute data from DrugBank for this use case, despite the option of using an academic license. When applying the tool to other use cases, however, the user is free to use whatever target-compound data source is available to them, be it web-based databases or own experimental data.
3. The authors explained the use case of CandActCFTR. It is not clear how the developed plugin will be implemented with new disease maps.
We thank the reviewer for drawing our attention to the missing instructions. We added a paragraph with a link to the step-by-step instructions on how to implement the tool for other disease maps.
4. The installation instructions for the plugin are not clear.
We have updated the documentation in the GitLab repository, which is now also linked in the manuscript, to include detailed instructions on how to install our plugin for own MINERVA instances.
5. The author mentions that the developed plugin is freely available in GitLab, but the GitLab link was not included in the manuscript. The only GitLab link provided is of starter-kit.
We would like to thank the reviewer for pointing out the missing GitLab link. We have now included the Link to the public project page in the abstract and “Supplementary Material” section.
6. Is the presented plugin allow the mapping of user-provided data? If yes, what type of data and how to upload this. Are there additional sources available to do that?
The aim of our plugin is that users can map their own data to relevant disease maps on MINERVA. We now added detailed instructions to the GitLab documentation on how to integrate and map own, user-provided data. In the manuscript, we now refer to the documentation specifically for the integration of user-provided data, however we refrained from including step-by-step instructions in the manuscript, as to not make it too manual-like.
Round 2
Reviewer 1 Report
This revision is appropriate.
Reviewer 2 Report
The authors have addressed all my comments in the previous review.